# Predicting SARS-CoV-2 Weather-Induced Seasonal Virulence from Atmospheric Air Enthalpy

**DOI:** 10.3390/ijerph17239059

**Published:** 2020-12-04

**Authors:** Angelo Spena, Leonardo Palombi, Massimo Corcione, Alessandro Quintino, Mariachiara Carestia, Vincenzo Andrea Spena

**Affiliations:** 1Department of Enterprise Engineering, Tor Vergata University of Rome, 00133 Rome, Italy; spena@uniroma2.it; 2Department of Biomedicine and Prevention, Tor Vergata University of Rome, 00133 Rome, Italy; palombi@uniroma2.it; 3Department of Astronautical, Electrical and Energy Engineering, Sapienza University of Rome, 00184 Rome, Italy; massimo.corcione@uniroma1.it (M.C.); alessandro.quintino@uniroma1.it (A.Q.); 4Department of Industrial Engineering, Tor Vergata University of Rome, 00133 Rome, Italy; mariachiara.carestia@uniroma2.it

**Keywords:** weather-related SARS-CoV-2 virulence, specific enthalpy of atmospheric moist air, temperature and humidity effects on COVID-19 outbreak, correlating equation, COVID-19 spread prediction risk scale

## Abstract

Following the coronavirus disease 2019 (COVID-19) pandemic, several studies have examined the possibility of correlating the virulence of severe acute respiratory syndrome coronavirus 2 (SARS-CoV-2), the virus that causes COVID-19, to the climatic conditions of the involved sites; however, inconclusive results have been generally obtained. Although neither air temperature nor humidity can be independently correlated with virus viability, a strong relationship between SARS-CoV-2 virulence and the specific enthalpy of moist air appears to exist, as confirmed by extensive data analysis. Given this framework, the present study involves a detailed investigation based on the first 20–30 days of the epidemic before public health interventions in 30 selected Italian provinces with rather different climates, here assumed as being representative of what happened in the country from North to South, of the relationship between COVID-19 distributions and the climatic conditions recorded at each site before the pandemic outbreak. Accordingly, a correlating equation between the incidence rate at the early stage of the epidemic and the foregoing average specific enthalpy of atmospheric air was developed, and an enthalpy-based seasonal virulence risk scale was proposed to predict the potential danger of COVID-19 outbreak due to the persistence of weather conditions favorable to SARS-CoV-2 viability. As an early detection tool, an unambiguous risk chart expressed in terms of coupled temperatures and relative humidity (RH) values was provided, showing that safer conditions occur in the case of higher RHs at the highest temperatures, and of lower RHs at the lowest temperatures. Despite the complex determinism and dynamics of the pandemic and the related caveats, the restriction of the study to its early stage allowed the proposed risk scale to result in agreement with the available infectivity data highlighted in the literature for a number of cities around the world.

## 1. Introduction

Temperature and humidity play a key role in the survival of a number of viruses, including severe acute respiratory syndrome coronavirus (SARS-CoV), Middle East respiratory syndrome-related coronavirus (MERS-CoV), and the influenza viruses [1,2,3,4,5,6,7], to which SARS-CoV-2 has recently been added [8]. In a previous work we noted that, although temperature or humidity cannot be independently correlated with virus viability, as demonstrated by the controversial outcomes from literature, a strong relationship between virus survival and the specific enthalpy of moist air, hereafter denoted as h, appears to exist [9]. In fact, according to the analyzed data, when the environmental conditions are such that the h value falls within the range of 50 and 60 kJ/kg of dry air, virus survival decreases dramatically. In contrast, data on h values, which may potentially be responsible for magnification of SARS-CoV-2 infectivity rather than its inactivation, have not yet been investigated extensively, despite an increasing but inconclusive number of studies that suggest geographic and climatic influences on coronavirus disease 2019 (COVID-19) outbreaks and spread [10,11,12,13,14,15,16,17,18]. Since human coronaviruses have generally shown a marked winter seasonality [19,20,21,22,23], the atmospheric h value at ground level may be tentatively used as a synthetic state property combining the simultaneous effects of temperature and humidity of moist air to evaluate the role of climate as a factor influencing the viability and diffusion of SARS-CoV-2, which is the scope of the present investigation. For the sake of realism, two preliminary statements are worth mentioning.

First of all, since any investigation combining climatic events with epidemiological events unavoidably crosses and merges two universes of empirical data, each of which is separately characterized by a high rate of uncertainty when compared to other fields of science or technology, the expected level of accuracy of the correlating results from this type of study is not particularly high. Moreover, such a tribute to natural complexity is a challenge that we have confronted in full awareness. This in order to acquire not only meaningful results, as solicited since February 2020 by the World Health Organization (WHO), which, among the current knowledge gaps, mentioned the “epidemic’s relation to seasonality” [24], but also the limitations and implications of their applicability.

Furthermore, since climatic conditions and especially enthalpy can considerably favor or hinder the spread of the virus according to known mechanisms (increase of survival in air and consequent increase in circulating viral load) and to others less known (effects on the host, on the respiratory tract, on comorbidities, etc.), the enthalpy analysis poses two different spatial/temporal kinds of problems:-At the stage of inspection, to establish an optimal time window sufficiently extended to allow: (i) ascertaining the onset of the epidemic, and (ii) acquiring epidemiological data sufficient for statistical purposes; but at the same time, not too prolonged so that: (iii) the community could still be considered as a nearly isolated system with respect to major external forcing independent variables (number of international travellers; degeneration into pandemic), and that (iv) inner forcing independent variables (government measures; anthropogenic factors influencing the spread of the infection) could not yet markedly act. In the present paper, for the time window of inspection, the order of magnitude of one month has been assumed.-At the stage of application: (i) to balance the enthalpy risk in the medium to long term with the constellation of other risk factors involved in the determinism of the pandemic, and (ii) to limit its use to geographic locations where the concept of seasonality has a sense in itself.

## 2. Materials and Methods

A study involving a comparison of the climate data from many cities around the world with and without significant community transmission was recently carried out by Sajadi et al. [25]. It was found that areas with higher COVID-19 progression were distributed roughly along the 30–50° North latitude corridor and experienced consistently similar weather patterns in terms of average air temperature around 7 °C and of average relative humidity (RH) of approximately 80%, as recorded by local airport weather stations 20–30 days prior to the first fatal COVID-19 cases. RH was defined as the ratio of the mass of water vapor to the mass of water vapor at saturation. However, the epidemiological threat was identified in terms of absolute numbers of either deaths or infected people, regardless of the percent incidence within the total population, and no attempt was made to define a unique climatic parameter that could be directly correlated with virus strength.

Based on these data, we proposed an elaboration by introducing the COVID-19 incidence rate (IR), which was calculated as the ratio between the number of ascertained cases and the total population of any of the cities included in the comparative analysis, and using the value of the thermodynamic potential enthalpy h to provide an effective synthetic representation of the thermo-hygrometric state of atmospheric air at ground level.

Subsequently, owing to the limited amount of data available for cities in which significant community transmission of COVID-19 occurred, the analysis was extended to a wider number of Italian provinces. This allowed us to propose an h-related risk scale, together with a temperature/RH chart, to simplify the prediction of non-negligible danger of SARS-CoV-2 virulence because of favorable weather conditions at the early stage of the epidemic.

## 3. Enthalpy Rationale

Enthalpy (from Greek ἐνϑάλπειν = to heat), denoted as H, is a state property introduced by the Dutch physicist H. Kamerlingh Onnes. It is defined as H = U + pV, i.e., the sum of the internal energy U of the system and the product of its pressure *p* and volume V, thus having the property that, when a process occurs at constant pressure, its variation is equal to the overall heat (sensible + latent) exchanged by the system during the process. Having the dimensions of an energy, enthalpy is measured in Joules (J) in the SI. In process analysis, environmental engineers normally use specific enthalpy h = H/m, i.e., enthalpy per unit mass, which is measured in J/kg. Since H, and then also h, is a state function, its change during any process depends only on the initial and final states, regardless of the transformation; additionally, its value is determined up to an arbitrary additive constant. In the following, we will deal with the specific enthalpy of the system represented by the atmospheric moist air, assuming, as usual, h = 0 at zero degrees centigrade and zero water content (dry air), and calculated as follows:h = c_a_t + AH (c_v_t + r)(1)
(RH%/100) = [*p*/p_s_(t)] × [AH/(AH + 0.623)](2)
where c_a_ and c_v_ are the specific heats at the constant pressure of dry air and of water vapor, which, around ambient temperature, can be assumed as equal to 1.006 kJ/kg °C and 1.86 kJ/kg °C, respectively; AH is the absolute humidity of moist air in kg_v_/kg dry air; t is the air temperature in degrees centigrade; r is the latent heat of vaporization of water at its triple point, equal to 2501 kJ/kg; *p* is the total pressure of the moist air, typically the barometric pressure related to the altitude above sea level in Pascal; and p_s_(t) is the saturated vapor pressure of water at temperature t in Pascal. The latter can be calculated from the empirical formula derived by Hyland and Wexler for the temperature range of 0 to 200 °C [23,26,27]:ln[p_s_(T)] = C_1_/T + C_2_ + C_3_T + C_4_T^2^ + C_5_T^3^ +C_6_lnT(3)
in which C_1_ = −5.8002206 × 10^3^, C_2_ = 1.3914493 × 10^0^, C_3_ = −4.8640239 × 10^−2^, C_4_ = 4.1764768 × 10^−5^, C_5_ = −1.4452093 × 10^−8^, C_6_ = 6.5459673 × 10^0^, and T is the absolute temperature in Kelvin degrees: T = t + 273.15. If the RH% is known instead of the AH, the AH value in Equation (1) to obtain the h value can be directly derived from Equation (2) as follows:AH = 0.623 [(RH%/100) p_s_(t)]/[*p* − (RH%/100) p_s_(t)](4)

## 4. Results

### 4.1. Elaboration of Available Literature Data

The elaboration of the results by Sajadi and colleagues is shown in Figure 1, in which the psychrometric chart [26] is superimposed on the distribution of the normalized data in order to show the related values of h and RH% of moist air. Notice that the psychrometric chart is based [26] on the absolute humidity (AH) instead of the specific humidity (SH), as originally done by the cited authors, AH and SH being defined as the ratio of the mass of water vapor to the mass of dry air in the moist air sample, and the ratio of the mass of water vapor to the mass of moist air in the moist air sample, respectively. Owing to normalization, the sizes of the circles were generally changed, here representing the total number of COVID-19 cases declared in each city with respect to the total population of each city; the red color was used to denote eight cities (namely Wuhan, Tokyo, Daegu, Qom, Milan, Seattle, Paris, and Madrid) where the most significant number of deaths occurred before 10 March 2020, which was the last day of data collection. Moreover, a number of circles were replaced by crosses when the percentage of COVID-19 incidence fell below 5 × 10^−4^%, which appeared to be almost negligible. However, the comparison with the representation originally done by Sajadi and colleagues illustrates that, at the beginning of the pandemic, most of the cities with the largest absolute numbers of infected people also shared the largest percentages of disease spread in the population.

As far as the h value is concerned, from Figure 1, it is apparent that the climatic conditions related to the largest incidence of COVID-19 are such that the average specific enthalpies span the range of 12–25 kJ/kg of dry air, which can be identified as the interval of the average h value corresponding to environmental conditions favorable to a surge in SARS-CoV-2 infectivity as known at the outbreak of the pandemic (consistent with Sajadi et al. [25] until 10 March 2020).

In contrast, the IR appears to have been practically negligible in cities whose average h value fell in the range of 50–60 kJ/kg dry air, which we recently identified as the environmental h interval with the lowest virus survival [9].

The relationship between h and the IR recorded in any of the cities with significant community transmission of the pandemic is shown in Figure 2. In order to identify the lower limit of the h range of atmospheric air corresponding to non-negligible spreads of COVID-19, we included the city of Moscow, which has one of the highest specific enthalpies among the densely populated cold cities that, according again to Sajadi and colleagues, recorded a negligible IR on 10 March 2020. Since the weather data were available from airport weather stations rather than urban weather stations, a reasonably average 1.5 °C increase was applied to the temperature data before calculating h in order to account for the well-known urban heat island (UHI) effect, as well as the notion that, in the winter, the UHI effect is generally less pronounced than in the summer [28,29,30]. In the same figure, a cubic interpolation curve with an R^2^ value of 0.643 (*p* < 0.001) is also displayed.

### 4.2. The Present Investigation

Although such a level of correlation could be considered satisfactory in the field of epidemiology, the limited data upon which the correlation is based cannot lead to the conclusion that the obtained result is sufficently robust to be used for safe predictions. Moreover, it must be noted that the eight cities shown in Figure 2 are not homogeneous at all in terms of social habits, urban texture, average age of the population, and availability of data. Therefore, a new wider data analysis was performed, using a procedure similar to the one described above, and was concentrated on relatively small geographical areas not too densely populated, as requested in order to obtain more significant results [31]. To do this, we calculated the IR in the first 40 days after the first reported case in 30 Italian provinces, excluding the city of Milan, which was already included in the study carried out by Sajadi et al. For any province, the h value was more accurately evaluated by considering the time evolutions of air temperature and AH during the 20–30 days prior to the first reported case [32], again introducing the temperature correction mentioned earlier to account for the UHI effect. Thus, instead of using the 10-day average values of temperature and humidity to calculate the average h, the 10-day average h of the atmospheric air was obtained as the median of the ten daily average h values, each determined by replacing the daily average temperature and AH in Equation (1). The results are listed in Table 1. In reality, the proposed set of Italian provinces, which can be considered as a representative sample of the whole country, since approximately 42% of the Italian people live there, is undoubtedly much more homogeneous than the set of cities considered by Sajadi and colleagues. Additionally, the climatic conditions of Italy span from those typical of the high-altitude mountains of Northern Italy (up to 46.5° N latitude) to those of the Mediterranean coastal areas (down to 36.5° N latitude). All the above allowed us to explore a significant wider range of both IR (0.02 to 1.18) and h (9.6 to 37.6 kJ/kg) values.

We performed a curve fitting analysis to find the best adaptation to our data. The relationship between the IR and the average h recorded during the 10-day observation period in the 30 selected Italian provinces is shown in Figure 3. The best-fit interpolation equation is presented as follows:IR (%) = 1.093 × 10^18^/[h^10.93^(e^177/h^ − 1)] − 0.15(5)

The observed R^2^ value of 0.798 (*p* < 0.001) is generally considered strong in the field of epidemiology. This reasonably enables us to infer that, at the beginning of the epidemic, up to nearly 80% of the variability in IRs (cumulative new cases per province population) was explained by h values. We can also note that, although other variables are certainly relevant in the determinism and dynamics of the epidemic, the h value seems to play a major role at the early stage of the epidemic (yet, potentially, at any stage), especially when its value ranges between 12 and 23 kJ/kg of dry air. This range corresponds to IRs > 0.5%, which is just below 50% of the peak rate (approximately 1.05%).

The shape of the curve confirms that the spread of COVID-19 started from climatic conditions having h values of atmospheric air spanning in the approximate range of 9–33 kJ/kg of dry air. Correspondingly, h values in the order of 0.03–0.3 kJ/kg of water vapor, together with AH in the interval from 3–9 g/kg of dry air, occurred in the selected provinces. The above conditions can be easily summarized by the assessment of a wet bulb temperature from nearly 0–12 °C, with a critical value around 4 °C, which is in particularly good agreement with the results from the study by Sahin et al. [15].

The asymmetry of the curve, even if weak, appears significant. While extinguishing slowly at the higher values of h, which seems to be complementary with our cited findings regarding significantly shorter coronavirus survival in the range of 50–60 kJ/kg of dry air [9], the suddenly sinking shape at the lower values of h clearly confirms that however high the RH value, the seasonal virulence of SARS-CoV-2 expires upon approaching 0 °C.

## 5. Proposal of an Enthalpy-Based Risk Scale

### 5.1. Risk Assessment

Apart from more specific considerations and regardless of the IR sampling time, a correlation between infectivity and both temperature and humidity of the outdoor environment appears to exist, allowing us to unlock the acknowledged methodological impasse that has continuously inhibited researchers from obtaining conclusive results. This enables us to propose an h-related risk scale for evaluating the danger of COVID-19 outbreak as a potential consequence of the persistence of weather conditions favorable for SARS-CoV-2, which is expressed in terms of seasonal virulence risk (SVR), as defined in Table 2.

In order to yeld an easy-to-use tool, a chart expanding upon the obtained h values in terms of coupled temperatures and RHs of the atmospheric air is depicted in Figure 4, where the corresponding levels of SVR are also marked. The dashed lines denote fields where the data rarely occur. It can be recognized that safer climatic and geographic conditions occur *with higher RHs at the highest temperatures*, *and with lower RHs at the lowest temperatures*. The latter statement is in good agreement with nearly all past observations in the literature [33,34,35,36], thus overcoming their apparent contradictions.

### 5.2. Validation

The proposed risk scale has been subjected to the following verifications. First, we examined the transmissions of COVID-19 in the eight cities mentioned in Figure 2, whose actual IR values and the related levels of risk have been compared with the SVR levels predicted according to the average h value in the 20–30 days prior to the recording of the first death or infection. The results are displayed in Figure 5, in which the intrinsically relevant data related to London and New York, whose IR percentages are 0.30 and 0.94, respectively, were also added for further comparison. It is apparent that, aside from the city of Daegu, the predicted enthalpy-based levels of the seasonal virulence risk appear to be in considerably good agreement with the available infectivity data.

As a second check, the proposed 12–23 kJ/kg of dry air range of h values corresponding to an average-to-high level of risk fits well with the results recently achieved by Ficetola and Rubolini [37], who assessed the effects of temperature and humidity on the global patterns of COVID-19 early outbreak dynamics from January–March 2020 throughout 121 countries around the world. In reality, the monthly average specific enthalpies of atmospheric air, calculated using the average temperature and AH data extracted from the WorldClim 2.1 raster layers [38] for the most severely hit countries included in their investigation, always span between 15–20 kJ/kg of dry air.

As further verification, the proposed scale resulted in good agreement with the outcomes of the study by Ahmadi et al. [14], who investigated the effects of a number of climatic parameters on COVID-19 spread throughout several provinces in Iran from 29 February–22 March 2020. From their results, a range of significant specific enthalpies roughly spanning from 11–28 kJ/kg of dry air can be ascertained.

### 5.3. Limitations

The above proposition is just an approximate method based on relatively limited data collected over various periods of time corresponding to the early stage of each local outbreak, with the aim of minimizing interference not only from healthcare restrictive measures, but also from any particular social behavior that may have been triggered by the growing fear of contagion. This inherently inhibits the survey of year-round data. However, within the limits of the adopted approach and the related warnings, the predictions appear sufficiently well-verified. This could motivate further developments that possibly consider other environmental forcing variables, such as the role of particulate matter (PM) and UV sunlight, or that deeply investigate to what extent our results could be applied to successive waves of a pandemic or to regions having nearly flat seasonal patterns of average temperature and RH values, such as the tropical ones. While beyond the scope of this work, such sensitivity analysis will be the subject of a forthcoming paper.

## 6. Discussion

To end the debate on whether or not climatic conditions can play an independent role—and, if so, to what extent—as a key modulating factor in COVID-19 onset and transmission [18,31,39], a number of other factors must be considered [24,40,41] in scenarios of not only high complexity, but also significant uncertainty, as mentioned at the beginning of this paper. The available number of cases and especially deaths in the total population are also influenced by the following factors: (i) the early detection of the pathogen; (ii) the number of investigations carried out, their statistical assessment, and possible under/overestimations [13,42,43]; (iii) demography in terms of population age and density [44,45]; (iv) urban texture, mobility, and social habits of the population [46,47]; (v) restrictions by local and national governments, such as quarantine and lockdown [35,48]; and (vi) medical care and susceptibility of the hosts [49,50,51]. These factors are almost entirely unrelated to h. On this topic, even in spite of possible correlations between some variables, the recent literature [52] has suggested that confounding factors, including some climatic ones [53,54,55,56,57], are nearly ready to be satisfactorily weighed. As an example, Nicastro and colleagues [58] recently stressed that outdoor living is much more common, even in winter, in Mediterranean countries than in North American states at similar latitudes, and that it is often anti-correlated with the extensive use of public and private indoor air-conditioning, which is suspected to be a mean of diffusion of contagions and could contribute to some extent to explaining why the SARS-CoV-2 epidemic has spread aggressively even at the very high enthalpy summer values of northern locations, like the states of Florida or Texas in the US, or in Israel, or Doha in Qatar. Similarly, the diffuse cloud coverage (especially at the highest solar irradiations) in tropical countries during specific periods of the year (the monsoon season in India, the rain seasons in northern Brazil, such as in Manaus, or hurricane seasons in central and tropical America) is considered [58] as a potential amplifier of the growth of contagions.

All of the factors listed above are generally expressed over periods of time of the order of several months to one year, which results in almost one order of magnitude higher than that of our survey (one month). This is the reason why, as discussed earlier, in order to mitigate the effects of the social and behavioral aspects as confounding factors, we restricted the collected data as far as possible to the first signs of the epidemic, namely the period from late January to early March. Moreover, to test the robustness of our estimations, we have qualitatively explored even more deeply the time preceding the outbreak of the pandemic. During the time when COVID-19 had not officially affected the populations and before stringent containment measures were implemented, the monthly statistical average h values from late 2019 to the beginning of 2020 in the three most affected global cities are reported in Figure 6, where the symbols denote the first recorded COVID-19 victim. It can be seen that the onset of the epidemic always followed a pronounced persistence into the average-to-high level of risk enthalpy area.

## 7. Conclusions

Based on the analysis of the first 20–30 days of the epidemic in a number of representative Italian provinces, we were able to determine a correlating equation between the IR of COVID-19 and h, as well as propose an h-related seasonal virulence risk (SVR) scale to predict the areas with a potential danger of non-negligible SARS-CoV-2 outbreak or magnification due to favorable weather conditions. Even though further investigations could corroborate the robustness of the discussed method in relation to pollution (mainly PM) and radiation (such as UV sunlight) effects, the presented findings confirm a relationship between SARS-CoV-2 distribution patterns and the specific enthalpy h of atmospheric air, as noted in our previous paper. Accordingly, a chart yielding the SVR value of any possible coupled temperature and RH value is provided. In particular, sites with specific enthalpies of atmospheric air in the range between 9 and 33 kJ/kg of dry air are classified as non-negligible risk areas, even though the actual virus virulence is contingent on a number of additional factors, such as social habits and mobility, the average age and density of the population, the healthcare settings, and the promptness of the government to impose quarantine or lockdown measures.

The proposed risk scale was verified by examining COVID-19 transmission in a number of cities all around the world highlighted in the literature, which demonstrated good agreement with the available early infectivity data. Furthermore, we showed that the epidemic outbreak typically occurred after an extended persistence into the environmental h range between 12 and 23 kJ/kg of dry air, which was classified as average to high SVR; and that, more generally, safer climatic and geographic conditions occur with higher RHs at the highest temperatures, and with lower RHs at the lowest temperatures.

Our method overcomes the methodological impasse caused by the fact that temperature or humidity cannot be independently correlated with coronaviruses viability, thus providing the scope for further investigation. The results show that, based on limited statistical weather data from any site having pronounced yearly temperature and RH patterns, the decision makers could predictively evaluate the season in which the corresponding h value falls in the domain of the ascertained starting preconditions for the possible onset or magnification and diffusion of COVID-19, and the period for which such a danger could occur.

## Figures and Tables

**Figure 1 ijerph-17-09059-f001:**
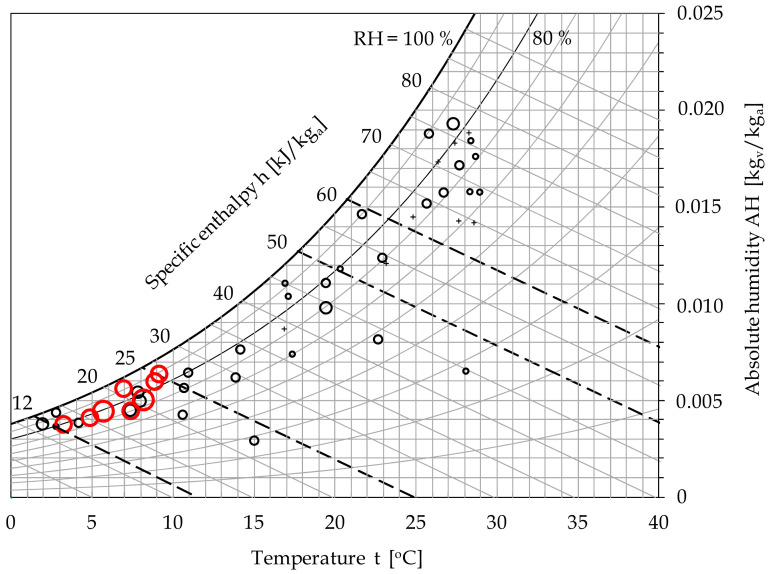
Elaboration of the infectivity data extracted from the study by Sajadi et al. [25].

**Figure 2 ijerph-17-09059-f002:**
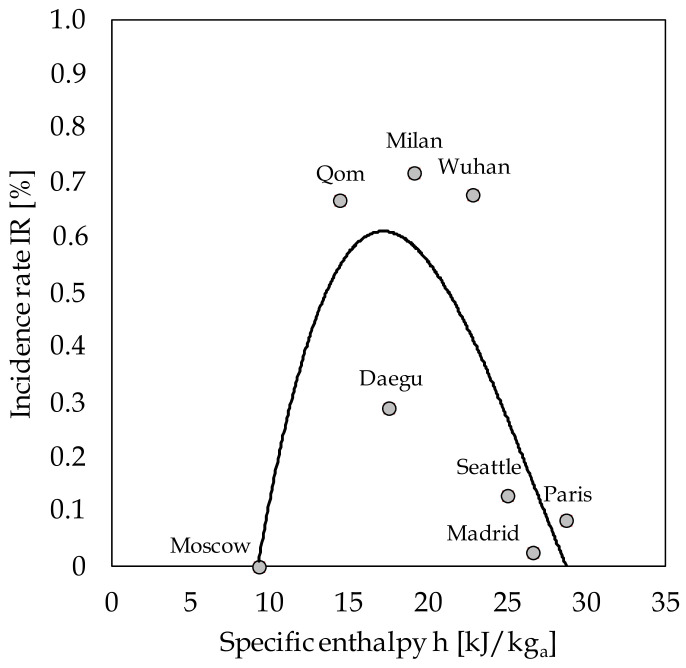
Interpolation curve incidence rate (IR) vs. specific enthalpy (h) for the seven considered cities, with the addition of Moscow (data extracted from the study by Sajadi et al. [25]).

**Figure 3 ijerph-17-09059-f003:**
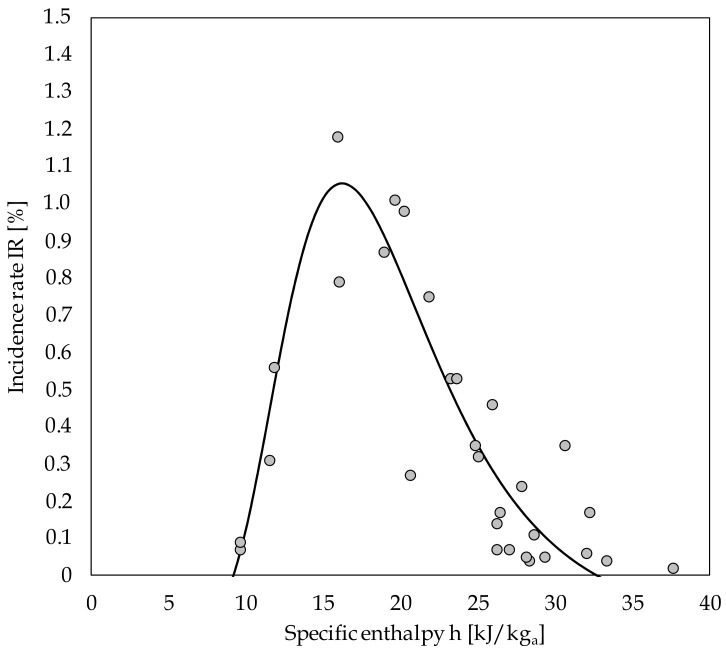
Interpolation curve incidence rate (IR) vs. specific enthalpy (h) for the 30 Italian provinces listed in Table 1.

**Figure 4 ijerph-17-09059-f004:**
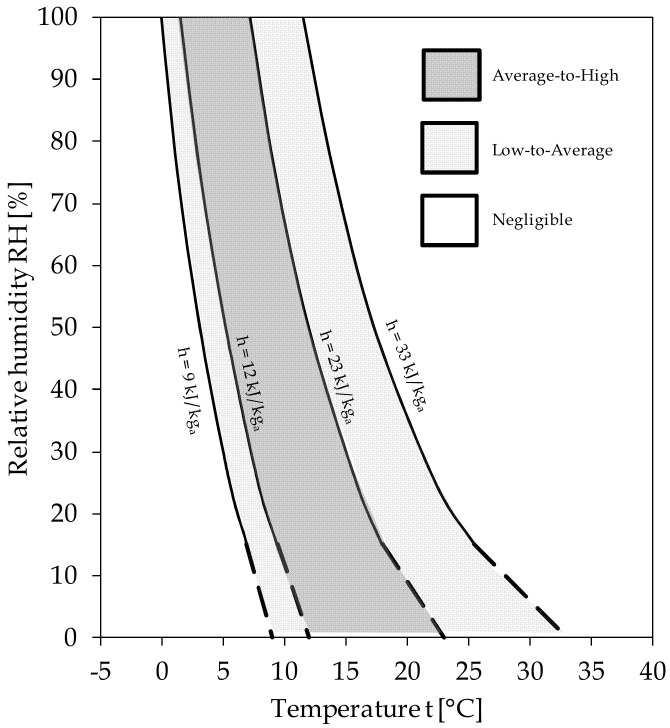
A chart of the suggested levels of seasonal virulence risk (SVR), expressed in terms of relative humidity (RH) and temperature of the atmospheric air.

**Figure 5 ijerph-17-09059-f005:**
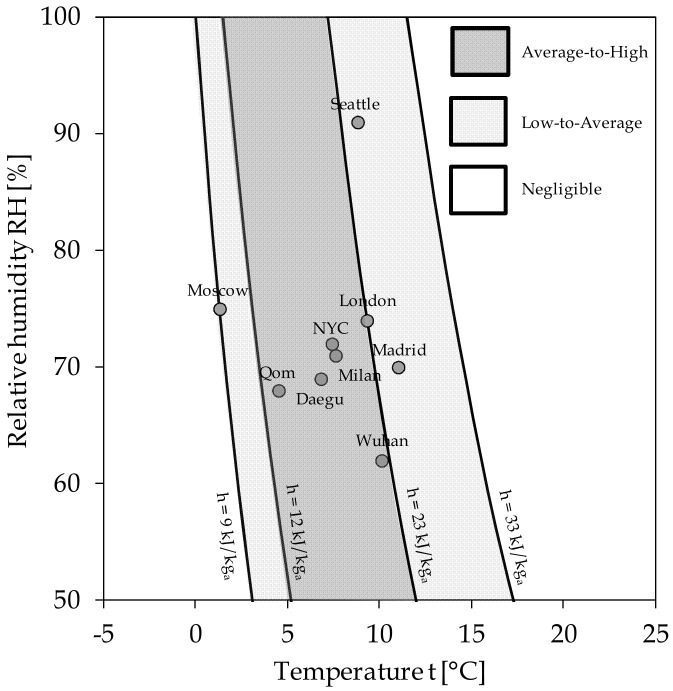
Verification of the cluster of ten global cities over the proposed chart of SVR levels.

**Figure 6 ijerph-17-09059-f006:**
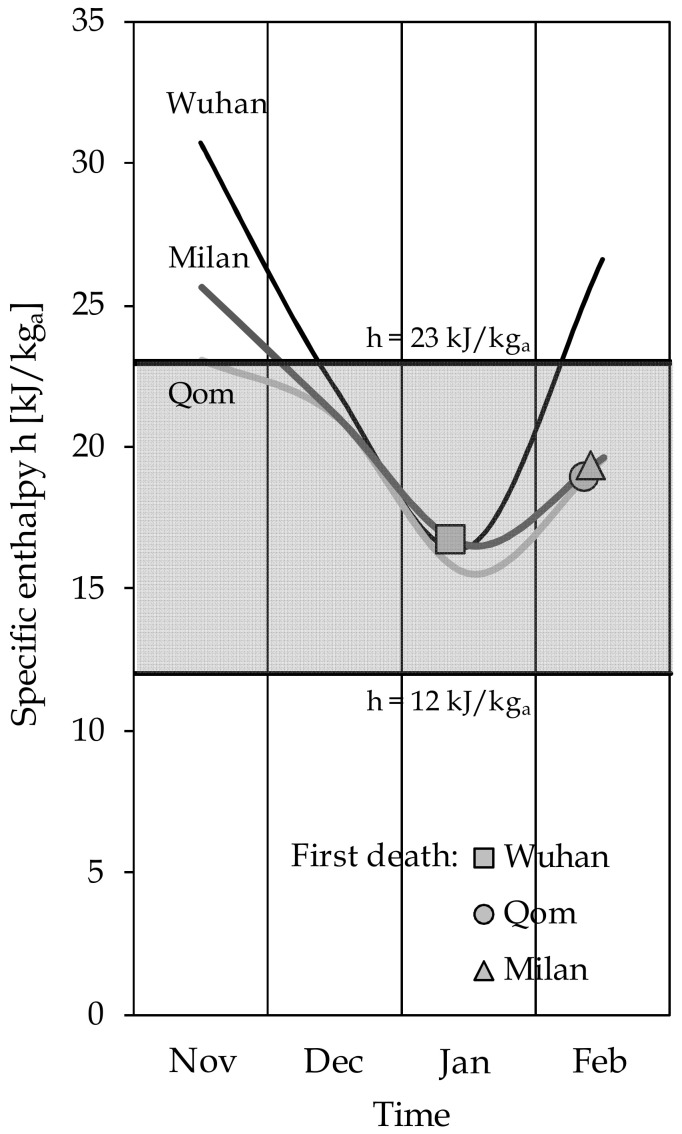
Time evolution of the monthly average specific enthalpy (h) statistical values from November 2019 to the time of the first COVID-19 related death in the most adversely affected cities.

**Table 1 ijerph-17-09059-t001:** Data from the 30 selected Italian provinces.

Provinces	Population	Cases After 40 Days	IR (%)	h [kJ/kg Dry-Air]
Alessandria	421,284	2248	0.53	23.6
Aosta	125,666	993	0.79	16.0
Bari	1,251,994	886	0.07	27.0
Bergamo	1,114,590	9712	0.87	18.9
Bolzano	531,178	1644	0.31	11.5
Brescia	1,265,954	9477	0.75	21.8
Brindisi	392,975	428	0.11	28.6
Cagliari	431,038	191	0.04	33.3
Campobasso	221,238	191	0.09	9.6
Cremona	358,955	4233	1.18	15.9
Firenze	1,011,349	1715	0.17	32.2
Genova	841,180	2918	0.35	30.6
L’Aquila	299,031	220	0.07	9.6
Latina	575,254	419	0.07	26.2
Lodi	230,198	2255	0.98	20.2
Napoli	3,084,890	1643	0.05	29.3
Palermo	1,252,588	299	0.02	37.6
Parma	451,631	2083	0.46	25.9
Perugia	656,382	950	0.14	26.2
Pesaro-Urbino	358,886	1919	0.53	23.2
Piacenza	287,152	2892	1.01	19.6
Potenza	364,960	162	0.04	28.3
Reggio Calabria	548,009	276	0.05	28.1
Roma	4,342,212	2714	0.06	32.0
Savona	276,064	654	0.24	27.8
Teramo	308,052	511	0.17	26.4
Torino	2,256,523	5985	0.27	20.6
Trento	541,098	3053	0.56	11.8
Trieste	234,493	821	0.35	24.8
Verona	962,497	3049	0.32	25.0

**Table 2 ijerph-17-09059-t002:** Proposed h-related risk scale.

Specific Enthalpy (*h*) Range	Level of Seasonal Virulence Risk (SVR)
*h* < 9 kJ/kga	Negligible
9 kJ/kga ≤ *h* < 12 kJ/kga	Low-to-average
12 kJ/kga ≤ *h* ≤ 23 kJ/kga	Average-to-high
23 kJ/kga < *h* ≤ 33 kJ/kga	Low-to-average
*h* > 33 kJ/kga	Negligible

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
