# Peer review of "Predicting SARS-CoV-2 Weather-Induced Seasonal Virulence from Atmospheric Air Enthalpy"

_ijerph, 2020, doi:10.3390/ijerph17239059_

Round 1

Reviewer 1 Report

This is a well-structured and well-written paper that builds upon previously published work by the same authors.

A plausible correlation is proposed between SARS-CoV-2 virulence and atmospheric air enthalpy.

The correlation is based on the derivation of atmospheric air enthalpy from temperature and humidity data acquired in several cities just prior to the onset of the current COVID-19 pandemic.

The analysis appears carried out convincingly, and the authors duly recognize that several possible confounding factors are to be further investigated.

The paper has an interdisciplinary nature, but it is directed to a readership of public health specialists.

As such, it would be appropriate for the authors to introduce the difference between relative, specific and absolute humidity values. Likewise, a description of the state function enthalpy would probably be helpful to most readers.

Minor comments

While the authors recognize that further analyses will be necessary, in the abstract they mention a “conclusive risk chart”. The adjective ”conclusive” is normally used to characterize something convincing or decisive. Thus, it appears that the authors possibly meant “final”.

On page 2, mention is made of a “specific humidity around 5 grv/kg”. The unit “grv” is not a standard one and it should be introduced. Moreover, should “gr” stand for gram, the appropriate symbol would be “g”. Should “v” stand for vapor, a subscript would be in order.

Author Response

The authors wish to thank the Reviewer for valuable comments and suggestions.

The manuscript has been widely modified accordingly, including additional changes useful to improve the clarity of the paper.

The answers to each point raised by the Reviewer are reported below.

Response to Reviewer #1

1.1 The paper has an interdisciplinary nature, but it is directed to a readership of public health specialists. As such, it would be appropriate for the authors to introduce the difference between relative, specific and absolute humidity values. Likewise, a description of the state function enthalpy would probably be helpful to most readers.

A.1.1 The definitions of Relative Humidity (RH), Absolute Humidity (AH) and Specific Humidity (SH) have been included in the revised manuscript, as well as the mention that enthalpy is a state variable allowing to combine the effects of temperature and humidity on the thermodynamic behavior of atmospheric moist air. See line numbers 80, 125-127

1.2 Minor comments: While the authors recognize that further analyses will be necessary, in the abstract they mention a “conclusive risk chart”. The adjective ”conclusive” is normally used to characterize something convincing or decisive. Thus, it appears that the authors possibly meant “final”.

A.1.2 We originally used “conclusive” as an opposite term to the “unconclusive” results generally complained in the literature, as we stressed in line number 40. In this frame, for the sake of the clarity, in the revised manuscript we substituted “conclusive” with the term “unambiguous”, see line number 26.

1.3 On page 2, mention is made of a “specific humidity around 5 grv/kg”. The unit “grv” is not a standard one and it should be introduced. Moreover, should “gr” stand for gram, the appropriate symbol would be “g”. Should “v” stand for vapor, a subscript would be in order.

A.1.3 Done, see line number 109.

Reviewer 2 Report

The article entitled " Predicting SARS-CoV-2 weather-induced seasonal virulence from atmospheric air enthalpy" by Angelo et al. studies the climatic effects on the virulence of SARS-CoV-2. As the author already mentioned, this work is an extension of the previously published work.

Major comments: Although the correlation with enthalpy seems a reasonable argument to make the exclusivity of the enthalpy as a function of virulence factor need to be validated in an isolated system. Theoretically, the virulence factor could have been calculated in an ideal controlled system while controlling the RH and temperature in the mentioned range with that of the model system.

Second, What is the author’s opinion on the recent surge of cases in the tropical and temperate zone? Do the temperature RH and enthalpy range analysis holds good to the tropical zone province like Florida, Texas, Israel, and the Indian sub-continent.

Third, It would be more insightful, if the author could have presented the enthalpy, RH/Temp over a time period, and see the infection rate for a particular province/country.

Minor Comment: Figure 1 has been published earlier. Authors are requested to take permission from the author/journal or change the figure and move it to a supplementary file if required to keep.

Author Response

The authors wish to thank the Reviewer for valuable comments and suggestions.

The manuscript has been widely modified accordingly, including additional changes useful to improve the clarity of the paper.

The answers to each point raised by the Reviewer are reported below.

Response to Reviewer #2

2.1 Major comments: although the correlation with enthalpy seems a reasonable argument to make the exclusivity of the enthalpy as a function of virulence factor need to be validated in an isolated system. Theoretically, the virulence factor could have been calculated in an ideal controlled system while controlling the RH and temperature in the mentioned range with that of the model system.

A.2.1 This is a correct theoretical concern. In the revised manuscript we stressed (§ 1-Introduction, see line numbers 63-73) that “the enthalpy analysis poses two different spatial-temporal kinds of problems:

  1. a) at the stage of inspection, to establish an optimal time window sufficiently extended to allow: i) to ascertain the beginning of the epidemic, and ii) to acquire epidemiological data sufficient for statistical purposes; but at the same time not too prolonged so that: iii) the community could be still considered as an nearly isolated system with respect to major external forcing independent variables (number of international travellers; degeneration into pandemic), and that iv) inner forcing independent variables (government measures; anthropogenic factors influencing the spread of the infection) could not yet markedly act. In the present paper, for the time window of inspection, the order of magnitude of one month has been assumed
  2. b) at the stage of application: i) to balance the enthalpy risk in the medium-long term with the constellation of other risk factors involved in the determinism of the pandemic, and ii) to limit its use to geographic locations where the concept of seasonality has a sense in itself.

2.2 Second, what is the author’s opinion on the recent surge of cases in the tropical and temperate zone? Do the temperature RH and enthalpy range analysis holds good to the tropical zone province like Florida, Texas, Israel, and the Indian sub-continent?

A.2.2 Well posed. Starting from the above answer A2.1 b), among the limitations to the use of the proposed methods, in the revised manuscript we dealth with this concern in four points:

  1. i) in the § 1-Introduction, see line numbers 58-73, when we say “limitations and implications of their applicability … the enthalpy analysis poses two different spatial-temporal kinds of problems ... to limit its use to geographic locations where the concept of seasonality has a sense in itself”
  2. ii) in the § 4.3-Limitations, see line numbers 270-274, when we say “This could motivate further developments that … deeply investigate to what extent our results could be applied to … regions having nearly flat seasonal patterns of average temperature and RH values, as the tropical ones. While beyond the scope of this work, such sensitivity analysis will be the subject of a forthcoming paper.”

iii) more in detail, in the § 5-Discussion, see line numbers 291-298, we have cited a recent Reference (n. 58) which we reassumed as “ … the extensive use of public and private indoor coarse air-conditioning, which is suspected to be a mean of diffusion of contagions and could contribute to some extent to explaining why the SARS-CoV-2 epidemic has spread aggressively even at the very high enthalpy summer values of northern locations like the States of Florida or Texas in the US, or in Israel, or Doha in Qatar. Similarly, the diffuse cloud coverage (especially at the highest solar irradiations) in tropical countries during specific periods of the year (the monsoon season in India, the rain seasons in the northern of Brazil as in Manaus, or hurricane seasons in central and tropical America) is considered [58] as a potential amplifier of the growth of contagions.”

  1. iv) in the § 6-Conclusions, we eventually stressed that our results can help decision makers only in countrieshaving pronounced yearly temperature and RH patterns”, see line numbers 335-336.

2.3 Third, It would be more insightful, if the author could have presented the enthalpy, RH/Temp over a time period, and see the infection rate for a particular province/country.

A.2.3 Given the high impact of anthropic factors (social distancing, healthcare intervention) it is likely that expanding the time frame of the analysis for the same region, will hide the effect of climatic conditions on the spread of the virus. Instead, we would like to expand this study to a wider number of countries, always referring to the very early stages of the outbreak, and before heavy safety measures were in place. This will be the aim of further research.

2.4 Minor Comment: Figure 1 has been published earlier. Authors are requested to take permission from the author/journal or change the figure and move it to a supplementary file if required to keep.

A.2.4 We reorganized the text in order to make unnecessary the use of the Figure 1, which was actually nearly redundant. In the revised manuscript the rearranged text spans between line numbers 85 and 133.

Reviewer 3 Report

This is a well written ms presenting an interesting association between climatic conditions and the spread of covid-19. However, the authors are extremely selective in choosing their data. Most of the data used is from the early phase of the pandemic when the time of introduction of the virus played a key role in the spread of covid-19. Also, the suggestion that the h factor influenced viral properties and thereby explains differences in epidemic is far-fetched. Human behaviour, e.g. time spent indoors, is also strongly influenced by climatic factors. Other factors, such as widespread nosocomial transmission in parts of Italy are not accounted for. 

When I plotted  two very severe epidemics (reaching "herd immunity" levels, i.e. over half the popuation got infected), namely the ones in Doha, Qatar and the one in Manaus, Brazil, onto Figure 5, I saw that these two cities should have had "negligible risk" !! But these data points were "wisely" omitted.

The hypothesis proposed might make some sense if indoor conditions and human behavior are taken into account.  It is up to the authors to show this.

Author Response

The authors wish to thank the Reviewer for valuable comments and suggestions.

The manuscript has been widely modified accordingly, including additional changes useful to improve the clarity of the paper.

The answers to each point raised by the Reviewer are reported below.

Response to Reviewer #3

3.1 However, the authors are extremely selective in choosing their data. Most of the data used is from the early phase of the pandemic when the time of introduction of the virus played a key role in the spread of covid-19. Also, the suggestion that the h factor influenced viral properties and thereby explains differences in epidemic is far-fetched. Human behaviour, e.g. time spent indoors, is also strongly influenced by climatic factors. Other factors, such as widespread nosocomial transmission in parts of Italy are not accounted for. 

A.3.1 Thank you for your comment and for giving us the opportunity to better clarify the scope of this work that was not sufficiently described.

Data from scientific literature showed the correlation of temperature, pressure and relative humidity and the ability of a virus to persist longer in the environment (indoor). These information can be extremely relevant at the early stage of the spread of the disease in the population: in fact, before any countermeasure is in place, the virus can easily circulate between the population and, especially for air transmitted diseases climatic conditions, can play a key role. Enthalpy showed to be an effective correlation parameter for that. Of course, the outbreak and evolution of an epidemic is strictly related to anthropic factors: however, this work supports the elaboration of a “risk map” of areas that can be more subject to the diffusion of COVID-19 in a given season, this information is a useful tool for decision makers to invest on preventive measures for an early detection of outbreaks in the future.

3.2 When I plotted two very severe epidemics (reaching "herd immunity" levels, i.e. over half the population got infected), namely the ones in Doha, Qatar and the one in Manaus, Brazil, onto Figure 5, I saw that these two cities should have had "negligible risk" !! But these data points were "wisely" omitted.

A.3.2 Well posed. Among the limitations to the use of the proposed methods, in the revised manuscript we dealth with this concern in four points:

  1. i) in the § 1-Introduction, see line numbers 58-73, when we say “limitations and implications of their applicability … the enthalpy analysis poses two different spatial-temporal kinds of problems ... to limit its use to geographic locations where the concept of seasonality has a sense in itself”
  2. ii) in the § 4.3-Limitations, see line numbers 270-274, when we say “This could motivate further developments that … deeply investigate to what extent our results could be applied to … regions having nearly flat seasonal patterns of average temperature and RH values, as the tropical ones. While beyond the scope of this work, such sensitivity analysis will be the subject of a forthcoming paper.”

iii) more in detail, explicitly concerning your request for clarification with respect to data from Doha, Qatar, and Manaus, Brazil, in the § 5-Discussion, see line numbers 291-298, we have cited a Reference (n. 58) which recently stressed that, as we say, “ … the extensive use of public and private indoor coarse air-conditioning, which is suspected to be a mean of diffusion of contagions and could contribute to some extent to explaining why the SARS-CoV-2 epidemic has spread aggressively even at the very high enthalpy summer values of northern locations like the States of Florida or Texas in the US, or in Israel, or Doha in Qatar. Similarly, the diffuse cloud coverage (especially at the highest solar irradiations) in tropical countries during specific periods of the year (the monsoon season in India, the rain seasons in the northern of Brazil as in Manaus, or hurricane seasons in central and tropical America) is considered [58] as a potential amplifier of the growth of contagions.”

  1. iv) in the § 6-Conclusions, we eventually stressed that our results can help decision makers only in countrieshaving pronounced yearly temperature and RH patterns”, see line numbers 335-336.

3.3 The hypothesis proposed might make some sense if indoor conditions and human behavior are taken into account. It is up to the authors to show this.

A.3.3 Of course, several concurring factors act simultaneously, as we mentioned several times in the paper. On the other hand, since February 2020 by the World Health Organization (WHO) mentioned the “epidemic’s relation to seasonality” as a factor to be inspected.

On this matter, in the revised manuscript we stressed:

  1. a) in the § 1-Introduction, see line numbers 58-72, when we say “limitations and implications of their applicability … the enthalpy analysis poses two different spatial-temporal kinds of problems ... - at the stage of inspection, to establish an optimal time window ... not too prolonged so that ... inner forcing independent variables (... anthropogenic factors influencing the spread of the infection) could not yet markedly act ... and - at the stage of application, to balance the enthalpy risk in the medium-long term with the constellation of other risk factors involved in the determinism of the pandemic
  2. b) that most of the concurring factors, see line numbers 299-303, are “generally expressed over periods of time of the order of several months-one year, which results of almost one order of magnitude higher than that of our survey (one month). This is the reason why … in order to mitigate the effects of the social and behavioral aspects as confounding factors, we restricted the collected data as far as possible to the first signs of the epidemic, namely the period from late January to early-March 2020”.

Again, while beyond the scope of our present work, such concerns will be the subject of a forthcoming paper.